# Q-LEARNING FOR REAL TIME CONTROL OF HETEROGENEOUS MICROAGENT COLLECTIVES

## ABSTRACT

The effective control of microscopic collectives has many promising applications, from environmental remediation to targeted drug delivery. A key challenge is understanding how to control these agents given their limited programmability, and in many cases heterogeneous dynamics. The ability to learn control strategies in real time could allow for the application of robotics solutions to drive the behaviour of microscopic collectives towards desired outcomes. Here, we demonstrate Q-learning on the closed-loop Dynamic Optical Micro-Environment (DOME) platform to control the motion of light-responsive *Volvox* agents. The results show that Q-learning is efficient in autonomously learning how to reduce the speed of agents on an individual basis.

## 1 INTRODUCTION

The ability to control the behaviour of agents at the microscale or smaller such has implications across fields such as nanomedicine (Hauert & Bhatia (2014)) and environmental remediation (Wang et al. (2019)), with possible agent types including micromotors, nanoparticles and bacterial cells. Exerting control at this scale remains challenging however, due in large part to the simplicity and limited programmability of typical microagents. In this work, an external optical control scheme is used control the microagents, here *Volvox* algae. Machine learning allows for the fine-tuning of the control to each individual *Volvox* in real-time. Through this, individual models can be learnt that enable optimal motion control, in this case learning how to alternate illumination and relaxation periods to stop the motion of individual *Volvox*.

Light is a powerful tool at the microscale, capable of forming and breaking bonds [Chen et al. (2018)], powering micromotors (Palagi et al. (2019)), and interacting with light sensitive organisms (Jékely et al. (2008)). Furthermore, the use of spatially structured light offers interaction with agents independently and in parallel (Palagi et al. (2019)), making it particularly well suited to the control of collective systems (Mukherjee et al. (2018); Izquierdo et al. (2018); Schmidt et al. (2019); Deng et al. (2018)). The dynamic nature of light also means that it can be combined with Q-learning to produce rapid and effective and closed-loop control outcomes. This was demonstrated by Muiños-Landin et al., with the use of tabular Q-learning on self-thermophoretic microswimmers to achieve navigation in a noisy, grid-like environment (Muiños-Landin et al. (2021)). The work presented here similarly uses tabular Q-learning to influence the dynamics of motile microscale agents using optical interactions, however in this case, each agent performs the learning independently, with significant heterogeneity present among the collective of agents owing to their biological nature. Furthermore, the learning and closed-loop optical control were here implemented on a low-cost, open source platform, demonstrating the power of this learning process even in instances with limited computational resources.

Optical control is enacted using the open source DOME platform, a light-weight device which combines digital light projection with microscopy to image a microsystem in real time and provide closed-loop localised light patterning. Given the limited computing power of the DOME, which operates on a Raspberry Pi computer, this work provides an exploration of the potential for the application of Q-learning algorithms in low computational resource environments.

Results show that tabular Q-learning allows us to learn how light may be projected onto *Volvox* algae in order to maximally reduce their velocity. The state and action space of a complex biological

system is simplified so as to run tabular Q-learning experiment, and the learnt values for individual agents used to achieve herding behaviour in living algae.

## 2 METHODOLOGY

This section introduces the experimental setup for light-based control of *Volvox*, the simulation environment, and Q-learning methodology applied both in simulation and reality.

### 2.1 OPTICALLY CONTROLLING *Volvox*

*Volvox* are a type of green microscopic algae that exhibit phototactic behaviour. They are multi-cellular organisms, with somatic cells that have flagella for locomotion and an eyespot for light perception. These cells allow the *Volvox* to move towards a light source (Ueki et al. (2010)). This phototactic response is adaptive, meaning that when a *Volvox* comes into contact with light its speed is typically reduced for around 2s before adapting to the new light environment and recovering previous velocity (Drescher et al. (2010)).

In this work, the light response exhibited by *Volvox* is used as a means to regulate the velocity of individual agents by providing spatially localised illumination. To overcome the adaptive nature of the response, illumination must be provided intermittently rather than as a continuous stimuli. Q-learning is therefore applied as a means to determine the optimum cycle length of illumination and relaxation for each agent that results in the largest velocity reduction. The *Volvox* used here were acquired from Blades Biological UK and are of the species *Volvox aureus*.

### 2.2 THE DOME

The experimental part of this work was performed using the DOME (Figure 1), an open source platform for the study and engineering of microagent collectives through spatiotemporal illumination (Denniss et al. (2020)). In this device, a closed-loop control scheme is established by linking a digital light processing unit to real time imaging and image analysis, enabling the optical micro-environment to be shaped around the evolving system dynamics. The DOME has a maximum projection resolution of $30 \times 30 \, \mu m$, and is thus well suited to illumination of individual *Volvox* agents, which are around 350–500 $\mu m$ in diameter.

### 2.3 Q-LEARNING FOR *Volvox* CONTROL

Due to inherent variability of living algae, in order to have an adaptable method of control of the *Volvox*, a reinforcement learning algorithm was required. Because the algae would be controlled using the DOME system, the learning algorithm could not be computationally expensive. Although this could have been circumnavigated by running the algorithm an external computer in communication with the DOME, this work aimed to explore the potential for implementing reinforcement learning in limited resource environments. Additionally, maintaining a self-contained computational set up allows for the possibility of operating the system in enclosed conditions, such as within an incubator for live cell study.

For this reason, tabular Q-learning was chosen, instead of more flexible alternatives such as deep Q-learning.

Tabular Q-learning has the restriction of needing a discrete action space and state space, but biological systems are inherently continuous. Due to this restriction, the action and state space were defined in a discrete way: The action space consisted on two actions, either to illuminate the *Volvox*, or not. The state space needed to represent the amount of light that a *Volvox* had received.

The *Volvox*'s speed is affected by the amount of light and darkness received. If the state space could be continuous, it would be defined by the amount of time (in milliseconds) that the agent had been illuminated and non-illuminated. Instead of measuring milliseconds, the measurement was discretised using the amount of frames. Since the number of states had to be finite, the number of frames of light or darkness could not grow infinitely. However, observation showed that after 10 frames of either illumination or darkness, the agent's behaviour did not change anymore. Because

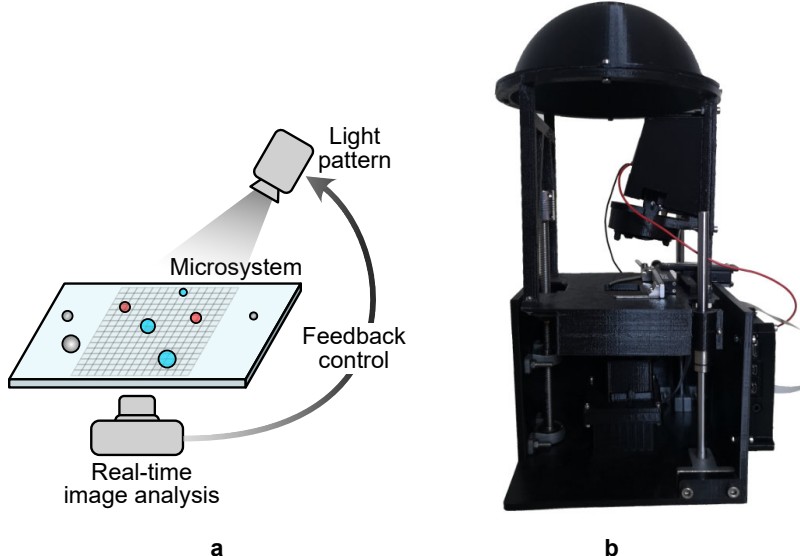

Figure 1: The DOME platform, shown schematically **(a)** and pictorially **(b)**. A digital light processing projection module is used to a controllable pixel grid of light. Real-time image analysis provided by standard light microscopy allow the light patterns to be shaped around the evolving dynamics of the microagent system to achieve closed-loop control.

of this, if an agent hasn't had a change in illumination for over 10 frames, it will be in the same state as if it had had the same illumination for 10 frames.

A state was therefore defined by the number of frames for which the agent was subjected to light, the number of frames for which it was not subjected to light, and the present light value. The present light value was necessary to distinguish between a state that had light on, then light off, and a state that had light off, then light on. Using this method, the total number of states was 242, which was the combination of possible frames on ($f_{on}$, between 0 and 10) and frames off ($f_{off}$, between 0 and 10) and the light value ($l$, either ON or OFF). Note that testing all combinations would not be possible in real time, as each trial is a real-world experiment involving a *Volvox* reaction. Table 1 shows some examples of states with their descriptions. The state index $S(f_{on}, off, l)$ was calculated as follows

$$S(f_{on}, f_{off}, l) = f_{on} + 11 * f_{off} + 121 * l. \tag{1}$$

| Description | $f_{on}$ | $f_{off}$ | $l$ | Index |
|---|---|---|---|---|
| The light was on for 15 frames, then off for 3 frames | 10 | 3 | OFF | 43 |
| The light was on for 7 frames, then off for 4 frames | 7 | 4 | OFF | 51 |
| The light was off for 4 frames, then on for 7 frames | 7 | 4 | ON | 172 |
| The light was off for 7 frames, then on for 4 frames | 4 | 7 | ON | 202 |

Table 1: Examples of states and their indices. The parameters $f_{on}$ and $f_{off}$ represent the number of frames for which an agent had light on and off respectively, while $l$ describes the present light value at a given point.

The reward for each state was calculated based on the agent's velocity ($v$) and acceleration ($a$) at that state. Since the goal was to minimize the magnitude of the velocity, rewards were given for agents with their velocity below a threshold, while accelerating agents were penalized. The direction of acceleration and velocity were not considered. Furthermore, because we wanted to minimize the number of light transitions (from on to off and vice-versa), states that had more frames on and off

would have higher rewards. The reward function $R(v, a, f_{on}, f_{off})$ was defined as

$$R(v, a, f_{on}, f_{off}) = \left\{ \begin{array}{ll} f_{on} + f_{off} & |v| < 0.05 \\ -5 & |v| \geq 0.05 \text{ and } a > 0 \\ -1 & |v| \geq 0.05 \end{array} \right\}. \tag{2}$$

At each step, the possible actions that could be performed were to turn the light either on or off for each agent, giving an action space of size 2. Given this, the Q-table was initialized as an empty matrix with $NumStates = 242$ and $NumActions = 2$, where each cell encodes the quality of choosing that action for that state. The Q-table was updated at each step of the learning as the agent explored the environment and different possible states. The Q-learning algorithm stored the previously chosen action ($action$), and the previous state ($s$), so as to update the reward at the next iteration.

### 2.3.1 VOLVOX SIMULATOR

The proposed learning methodology was refined in simulation before use in reality. To this end, an agent-based *Volvox* simulator was built in Python to perform rapid iterations on the control algorithms. This simulator replicates the way in which *Volvox* behave in response to light, and was designed such that all code developed was also suitable to run on the DOME platform. *Volvox* agents were modelled based on three assumptions from observation and literature:

- Agent velocity is reduced for period of time when coming into contact with light.

- After a period of time in contact with light, agent velocity recovers.

- The duration of the aforementioned two time periods vary from agent to agent.

The simulated agents follow a straight line, with a probability of them changing direction. This is to replicate the randomness of the *Volvox* movement. In both the simulator and real world experiments, the passage of time was broken up using the number of elapsed camera frames, allowing an otherwise continuous measurement to be discretised. Since each *Volvox* reacts to light in a different way, there exists are a pair of values for the number frames with light on $f_{on}$, and number of frames with light off $f_{off}$ that if repeated continuously, will keep the agent at its minimum speed. To model this light responsive behaviour of the *Volvox*, a light accumulator model was developed. The emulated agent had two local variables, in which the amount of light ($a_L$) and darkness ($a_D$) were stored. These variables were bounded between the values of 1 and 20, and increased exponentially with every new frame of light or darkness. The choice of having an exponential increase was to reflect the fact that at each frame of light that an agent received, the agent would have more capacity to absorb the light, because it would adapt to its new illumination environment. If any of these variables went above the maximum value, it meant that the agent would no longer react to that impulse. In the following equation, $t_L$ indicates the number of consecutive frames of light, and $t_D$ indicates the number of consecutive frames without light.

$$a_L(t_L) = e^{\lambda_L * t_L}$$
$$a_D(t_D) = e^{\lambda_D * t_D}$$

The parameters $\lambda_D$ and $\lambda_L$ indicate the rate at which an agent stops reacting to darkness or light, respectively. These values were calculated based on the number of required on and off frames for the agent to stop.

$$\lambda_L = \ln(20) * \frac{1}{f_{on}}$$
$$\lambda_D = \ln(20) * \frac{1}{f_{off}}$$

The code used for this simulator is publicly available online at bitbucket.org/hauertlab.

# 3 RESULTS

## 3.1 SIMULATION

Initially, the *Volvox* simulator described in Section 2.3.1 was used to develop and test a learning algorithm for reducing agent velocity.

### 3.1.1 SINGLE AGENT CONTROL

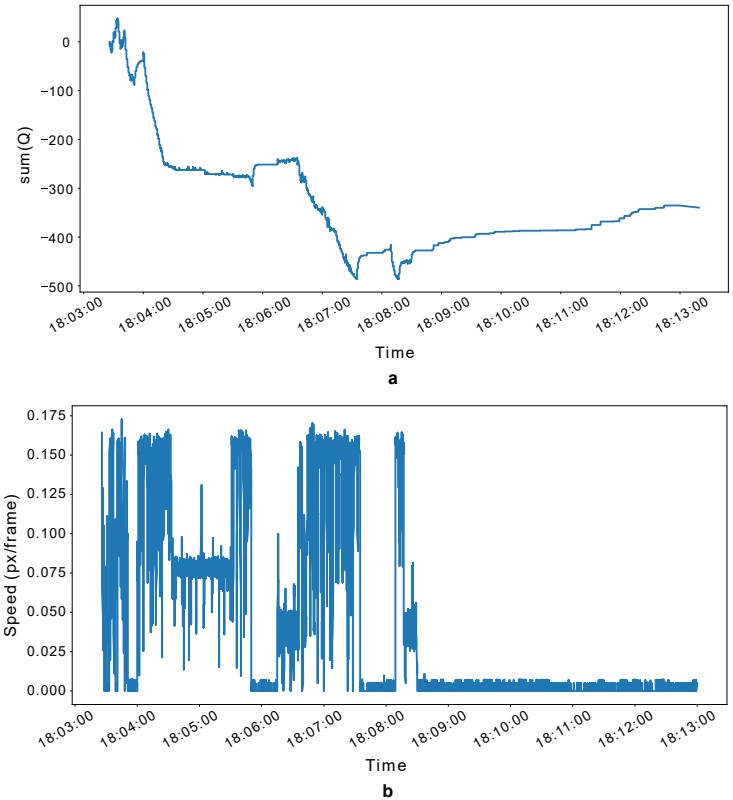

Figure 2: **(a)** Q-table sum converging after 10 minutes for one simulated *Volvox*. **(b)** Speed decreasing over time for one simulated *Volvox*.

The Q-learning algorithm was initially run on the simulator for a single agent, attempting to reduce agent speed as much as possible for the longest amount of time. The total duration of the experiment was of 10 minutes, representing a typical trial in reality. The simulated agents were programmed to stop when they had received 4 frames of light, and then 3 frames without light. The goal of the Q-learning algorithm was to learn this sequence of actions in order to stop it. The sum of the Q-table over time (Figure 2a) shows that initially, many of the rewards were negative because the agent was penalized. After 5 minutes however, the table values stagnate as the agent has discovered which states will yield the largest rewards. Accordingly, Figure 2b similarly shows that the agent has a variable speed until 5 minutes, at which point the speed stabilizes at low values. The learnt Q-table was analyzed to understand the best actions for the agent, as well as the actions that were chosen most frequently in the last 1000 actions.

### 3.1.2 MULTI-AGENT CONTROL

The same learning process was repeated for a system of multiple emulated *Volvox* agents, replicating a typical experiment. The plot in Figure 3a shows how the speeds of all the detected agents is reduced over time. Consistently across agents, velocity is ultimately reduced after a period of variation during which the learning process occurs.

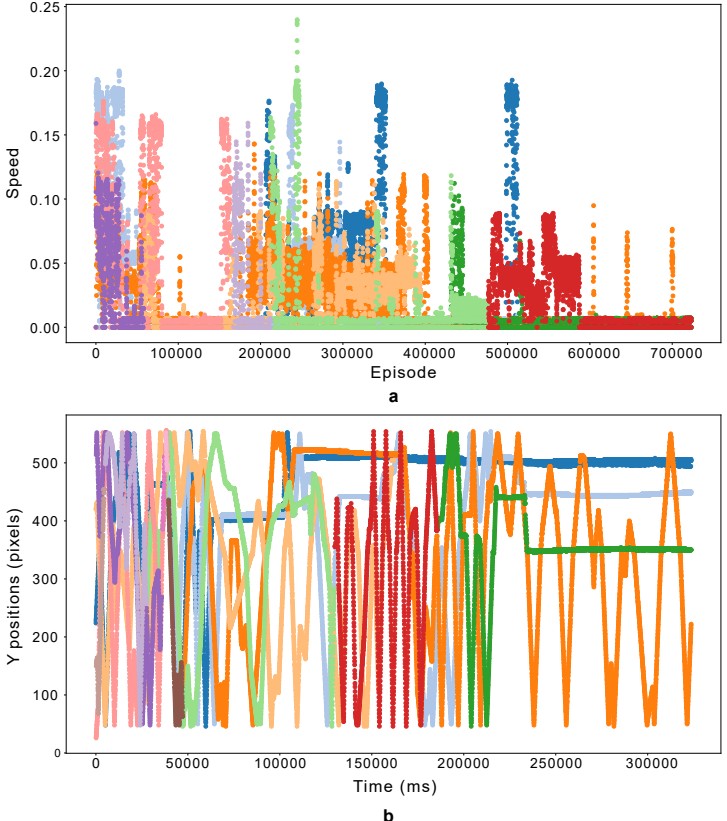

Figure 3: **(a)** Speeds of multiple agents over time, as Q-learning attempts to stop them. Speeds are reduced over time because the learning algorithm discovers optimal parameters for each agent **(b)** Y positions over time of emulated agents during herding, showing that 3 agents stop moving in positions where $Y \geq 328$. For both plots, each colour represents a different agent. It can be seen that one agent (orange) has not yet learned the illumination pattern required to stop moving.

The velocity control demonstrated above was then used to explore the possibility of 'herding' the *Volvox* by attempting to gather agents at the lower part of the screen where $Y \geq 328$. The agents positioned in this lower half were controlled using the learning algorithm in an attempt to prevent them moving out of the area, while the other ones were not illuminated, allowing them to move freely. Figure 3b shows that over time, 3 of the agents move to the parts of the screen with high Y coordinate. For the time point shown, the learning algorithm had not yet learnt how to control the agent plotted in orange, hence the position moves up and down the screen continuously while other agents are controlled and kept at the correct position.

## 3.2 EXPERIMENTAL VALIDATION

The algorithms developed in simulation were then implemented on the DOME for experimental validation with real *Volvox* agents, with the aim of showing velocity reduction and, if possible, herding. First, three runs with no Q-learning were performed to provide a comparison point. The movement of *Volvox* was initially observed under no illumination, then under continuous localised illumination. Following this, a blinking illumination experiment was run in which light was provided intermittently at $f_{on} = 1$ and $f_{off} = 1$ to reduce the degree to which *Volvox* were able to adapt to the light without using a complex learning algorithm. For the continuous and blinking illumination, localised light was provided to the *Volvox* agents positioned in the lower half of the sample, or in terms of image coordinate system, where $Y \geq 328$. This aimed to recreate the conditions that led to a herding outcome in simulation. Following the previous experiments, the Q-learning algorithm was run on the *Volvox*, a video of which can be found at youtu.be/Uep5J6RIGHM.

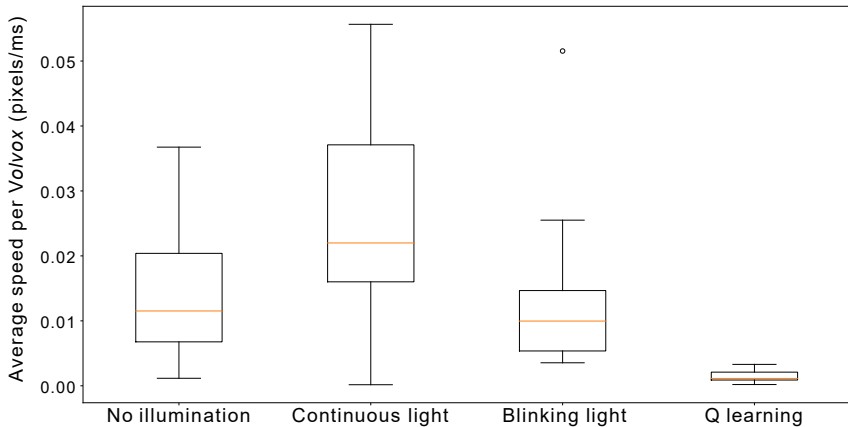

Figure 4: Average speeds in each of the experimental conditions, showing that using Q-learning, there is a lower average speed and less variance in the Y position. For the blinking light condition, an outlier can be seen, represented by a circular point.

The speed of the *Volvox* was compared across the three conditions to understand how effectively each method achieved velocity reduction. The box plot in Figure 4 was created by averaging the speed of each of the detected agents over each of the experiments. Except in the case of no illumination, only *Volvox* speeds from the illuminated portion of the sample (continuous or blinking) were considered. The plot shows that the Q-learning algorithm maintains the *Volvox* at a lower speed than the other strategies with significantly less variance. Despite blinking light having a low average speed, there is an outlier that moves at more than 0.05 px/ms, meaning that the algorithm is not good enough to stop all agents. A t-test was performed comparing the Q-learning speed values with each of the other conditions, all three of them showing that the difference was significant ($p < 0.0003$). Similarly, a t-test was done comparing the speeds of the continuous illumination condition with the others, all of which showing that the difference was significant ($p < 0.03$). In all of these conditions, it was possible for agents to stop moving, despite not having changes in illumination, due to the randomness of biological systems.

To better understand the qualitative behaviour of learned strategies, two agents that had been tracked over a long time period were analysed. For both agents, the Q-tables (Figures 5a and c) converged, with the sum of the table stagnating after some time, and agent speeds were kept under 0.05 px/ms during the whole detection. The first part of the corresponding Q-table for agent A (Figure 5b) shows that the reward is maximum when the light was on for 3 frames, and then off for 4 frames. For agent B however, the Q-table outcomes (Figure 5d) differ from that of agent A, suggesting that each *Volvox* reacts to light in a different manner.

During the learning phase, the algorithm tried different combinations of actions depending on the state of the Q-table at that moment. In Figure 5e, eight consecutive frames from the camera are shown. In all of them, two agents are detected, but each is illuminated at a different rate to keep the speed as low as possible, with continuous analysis of the speed to update the rewards of each state. The agent at the top left starts illuminated ($t = 0s$), then light is turned off for 1 frame (until $t = 0.3s$), then back on for 2 frames (until $t = 0.9s$), then off for 2 frames (until $t = 1.5s$), on for 1 frame, and off again. The agent at the bottom right also starts illuminated ($t = 0s$), then light is turned off for two frames (until $t = 0.6s$), then is turned on for three frames (until $t = 1.5s$), then back off for two more frames (until $t = 2.1s$).

## 4 DISCUSSION

Q-learning is suited in an unknown environment, and can be used to understand the best way to control microscopic agents in a way that is independent from the agent type and its characteristics. This means that, although *Volox* algae are employed here as a model microagent, the tools developed could be adapted to suit other stimuli responsive agents. As with most organic systems, and many inorganic, large degree of heterogeneity exists in the stimuli-responses of *Volox* agents even within

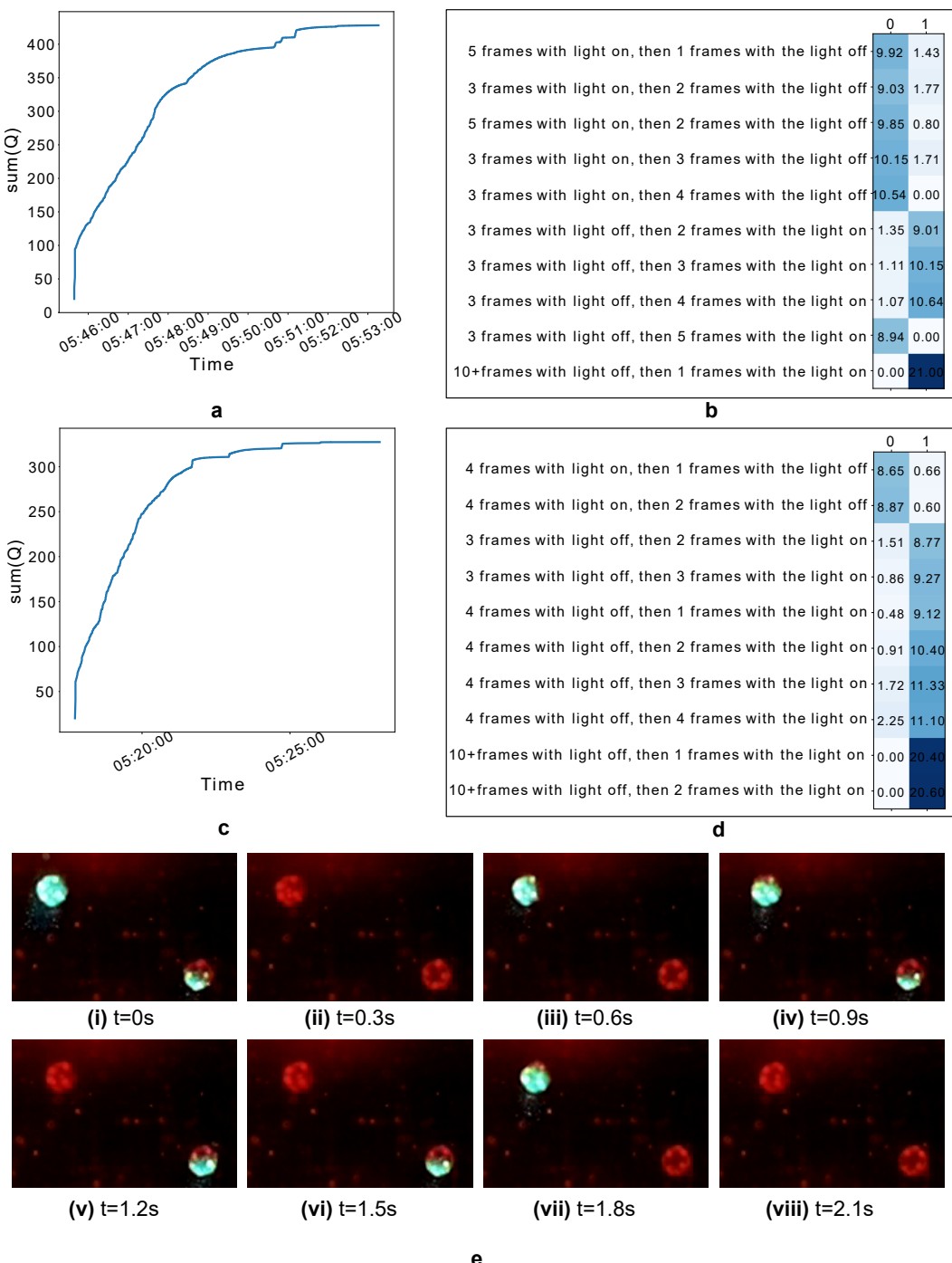

Figure 5: **(a)** Evolution of the sum of the Q-table for agent A, showing stagnation. **(b)** Partial Q-table values for agent A, showing that after 3 frames off, then 5 on, the best action is to turn the light off again. **(C)** Evolution of the sum of the Q-table for agent B, showing stagnation. **(d)** Partial Q-table values for agent B, showing that after 4 frames with light off, then 4 frameswith light on, the best action is to keep the light on. **(e)** Two *Volvox* illuminated at different rates using Q-learning.

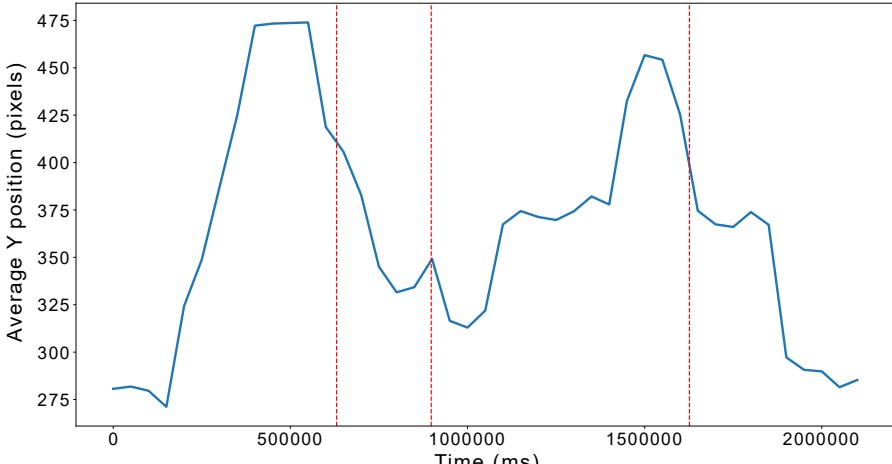

Figure 6: The average Y position of all *Volvox* agents over time the Q-learning algorithm alternates between illuminating the top and bottom of the screen. Red dashed lines indicate the time point at which the illumination half was switched.

the same population. For this reason, learning individually tuned parameters is powerful in achieving precise control.

In both simulation and experiment, Q-learning proved a successful strategy for reducing the velocity of emulated and real *Volvox* agents respectively. The results of the Q-tables differ between simulation (Figure 2a) and experiment (Figures 5b and d) in that values are larger in real-world experiments than in simulation. This is possibly because the simulator assumed that stopping the *Volvox* would be more difficult than it turned out to be, meaning that agents were less likely to be rewarded. Despite this, both simulated and real results found that agents do not only have one combination of light on and off that is valid. Instead, there are many combinations that allow to keep a low speed, and Q-learning successfully learns these. In the experimental section, it was found that Q-learning provided the most efficient strategy for slowing the *Volvox* when compared to standard continuous or intermittent illumination patterns. This was demonstrated by the lower average speed, and smaller variance in speeds seen in Figure 4.

In addition to the ability to regulate velocity, herding of agents into a particular area was also explored. In simulation, this was found to work well, with the Y positions of all but one agent being inside the chosen half by the end of the control period (Figure 3b). In experiments, preliminary results suggest that the same may be possible. Figure 6 shows the average position for a collection of *Volvox* agents over time, where the illuminated section of the space was switched three times during the experiment. In all cases, the switch occurred when most or all agents had moved to the illuminated region. This experiment suggests that using Q-learning, light could be used to gather agents in an area of the sample despite not being able to directly control their direction. However, due to the small agent number (4-5) and the large variance in the natural movement of *Volvox*, even in the absence of light, further experiments are required to verify this outcome.

The potential to control an entire microagent collective in parallel could also allow for exploration of swarm behaviours and control strategies at the microscale. Broadly, a swarm system is one in which agents are able to collectively perform actions that are beyond the capabilities of an individual, typically facilitated through local interactions (Brambilla et al. (2013)). Unlike in macroscale swarm engineering, microagents cannot be straightforwardly programmed with interaction rules, rather agents must typically interact through physical means such as chemical signalling. This requires the design and fabrication of highly complex agents, something that can be costly and time-consuming. Given this, the development of generic microswarm control strategies could be crucial in informing the design of these intelligent micro/nanoagents, efficiently directing the production process to best suit a given application.

Overall, this work suggests that tabular Q-learning is a useful tool to efficiently learn microagent control strategies in real-time on platforms with limited computational resources, as it is not as computationally expensive as other more complex learning algorithms, and is suited to an unknown environment. The use here of the DOME as a low-cost, open source platform is also significant in widening accessibility to similar control techniques.

## 5 FUTURE WORK

The primary goal of this project was to demonstrate the potential of using Q-learning to control individual agents within a complex, living biological system in real time. In line with this, a control strategy developed using tabular Q-learning was found to be effective compared to non-learning baselines in regulating the motion of these biological agents. In future work, comparison with alternative learning-based algorithms would be informative in optimising the control process. Furthermore, due to the on-board, lightweight nature of the computational strategy developed here there is the potential to apply similar schemes to live mammalian cell environments through operation inside an incubator environment, or even as a miniaturised wearable medical device. This could allow exploration of learning based control for cell collectives such as tumours or healing wounds.

ACKNOWLEDGMENTS

This work was supported by an EPSRC DTP scholarship (A.R.D) and the EPSRC TAS pump priming fund (A.R.D, S.H., T.E.G.).

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
