# OpenReview forum: "Q-learning for real time control of heterogeneous microagent collectives"
_ICLR.cc/2022/Conference — ICLR 2022 Submitted_

### Official Review · Reviewer_GPp7 · 2021-10-26

**Correctness:** 3
**Technical Novelty And Significance:** 2
**Empirical Novelty And Significance:** 2
**Recommendation:** 3
**Confidence:** 2

**Main Review:**

 Strength:
* The paper is  clear to understand on a high level, given the application domain. The authors motivate the use case well, justifying the case for applying Q learning for this task.
* Its (to the best of the reviewers knowledge)a novel application of Reinforcement Learning

Weakness:

Quality of work

 *  This is an area for improvement. The level of contribution could be improved. An example would be adding other baselines apart from Q learning. The choice of baselines, which seem like hardcoded policy(random, blinking etc), might not be a fair comparison to the Q learning agent.

Other comments:
 * "The Q-learning method requires having a finite number of states and actions, and stores a Q-table
 with as many rows as states, and as many columns as actions "-> It'll be good to clarify this as tabular Q learning. While this is correct, recent advances typing Deep Learning with Reinforcement Learning have allowed networks to generalize from a large state space which is infeasible to be stored in tabular format. In this context,  an interesting follow up would be to combine the pixel values of the microscopic image with the structured state information.

* While it intuitively makes sense on penalizing velocity, acceleration, perhaps more background can be provided on the motivation behind this. Does the direction of velocity also matter or only the absolute magnitude is penalized?

* How well does the dynamics of the simulated environment map to the real world? The approximation seems reasonable, but it would seem that an agent would be able to learn an optimal policy, ie there is no stochasticity, randomness that would add complexity to the environment.

*  Is it possible for the speed to be 0 with continuous illumination?

* If the agent stays in either "light"or "dark" state for more than 10 frames at a time, how will that be characterized? Is that a feasible scenario?


**Summary Of The Paper:**

The given work discusses the use of Q learning to control the motion of a light responsive Volvox agent. They also develop a simulation environment, providing an empirical estimate of  the dynamics of the system. They evaluateon both single and multi-agent control. The proposed tabular Q learning agent was able to achieve superior performance(ie slower speed in this setup) compared to the baselines proposed.

**Summary Of The Review:**

The paper provides the application of Tabular Q based learning on a a novel application. The author's also develop a simulation of the environment for to test this policy. The paper is reasonably well written, clearly organized. However, the contributions, significance and empirical evaluation are limited. Tabular Q learning is a well studied concept. Only a single learning algorithm is evaluated, while the choice of baselines could be expanded to other agent types.

---

> ### Author Response · Authors · 2021-11-22
> **Response to Reviewer GPp7**
>
> Comparable baselines would indeed be informative to this project and will be investigated in future work. Here, our main objective was to gauge the effectiveness of tabular Q-learning strategies in controlling a complex biological collective in real time. As such, our primary concern during the experiments was to test whether this learning algorithm performed well compared to the hardcoded baselines. Moving forward, will we aim to optimise this control strategy through the testing of alternative machine learning techniques.
>
> The questions presented in other comments provided helpful points in need of clarification, which hopefully are clearer in the updated paper version, for example the specification of tabular Q-learning and that only magnitude of velocity was considered.

---

> > ### Comment · Reviewer_GPp7 · 2021-11-24
> > **Thanks for answering question**
> >
> > The motivation is more clear to me. With this, I've decided to keep my score.

---

### Official Review · Reviewer_BtTc · 2021-11-02

**Correctness:** 4
**Technical Novelty And Significance:** 2
**Empirical Novelty And Significance:** 3
**Recommendation:** 6
**Confidence:** 2

**Main Review:**

The main strengths of this paper:

(1) It applies machine learning to real world problems. Such inter-discipline studies are highly encouraged imho, since they demonstrate the real power of machine learning outside standard benchmarks and theories.

(2) It demonstrates that the learned Q table can regulate the motion of the algae significantly better than human designed baselines.

(3) Experiments are conducted in both  simulated and real world environments.

The main weakness of this paper:

(1) Overall presentation of this paper can be improved.  For example, the state space of the problem is not clearly described and involves some confusing details. The authors seem to use past light on/off history as the state space. From my understanding it is equivalent to concatenating applied actions (turning on light or off)  as the state space. Is that the case? Also, the authors write "Additionally, if no change in state was detected after 10 consecutive frames in either light or darkness, the state would no longer change." What does this mean? How does it change the state space?

(2) Given how the authors design the state space, I think there is a more continuous way to formulate the state space: the micro-agent's velocity as the state, and on-off of light as the action. In this way we will have an alternative formulation. The authors can still discretize the velocity space if they want to use Q-table, but they can also use DNN to represent the Q network. Has this been tried?

(3) No videos etc to demonstrate the real world experiment. Videos will help the readers, who may not be from this field, to understand better.


**Summary Of The Paper:**

In this paper, the authors use machine learning to control the velocity/motion of a type of microorganisms, the Volvox algae. The algae's velocity will react to light in a non trivial, adaptive way,  and the authors use Q learning to regulate the algae's velocity. Experiments are conducted  in both simulator and real world. The authors demonstrate that after learning, the Q table method can reduce the motion of this micro-agents to be almost stationary, significantly better than baselines.

**Summary Of The Review:**

In summary, the authors apply Q-learning to solve a real world control problem for microorganisms. The approach is inter-disciplinary, though the method  (Q table) is well known. The authors demonstrated compiling results in both simulator and in the real world.

---

> ### Author Response · Authors · 2021-11-22
> **Response to Reviewer BtTc**
>
> It does indeed seem a good idea to formulate the state space in a more continuous manner as described. For this work in particular, a discrete formulation was chosen as it seemed to relate the most simply to the way the system had been set up at that point, with discrete time points (frames). Moving forward however, this approach would be interesting to adopt. The first point regarding specific clarifications has been addressed in the updated text, and a video has been made available, linked in text, to better illustrate the nature of the experiments.

---

### Official Review · Reviewer_QBsR · 2021-11-05

**Correctness:** 3
**Technical Novelty And Significance:** 2
**Empirical Novelty And Significance:** 2
**Recommendation:** 3
**Confidence:** 4

**Main Review:**

In the tabular Q-learning instantiation, the state space was defined by the amount of consecutive light that an agent had received measured by the number of frames instead of time. The reward is based on agents’ velocity and acceleration The Q-table was initialized as an empty matrix with 242 states and 2 actions, where each cell encodes the quality of choosing that action for that state.

The paper presents an application of tabular Q-learning to a new and interesting environment (DOME). As such the main contribution is the formulation of states, actions, and rewards to the well-known Q-learning algorithm. As such I am concerned that the scope of the paper is suitable since the contribution mostly focuses on the environment rather than the learning algorithm itself.

Comments that could help improve the paper:

- The paper also presents a simulator. It would be good if it were made available to the public.
While the abstract describes the learning process for a collective, the learning actually works on a per-agent level. It would help if this could be clarified earlier on.

- To strengthen the paper, comparisons to other competitive algorithms and baselines are needed. Currently, no comparison algorithms, no comparable baselines, and no ablations are provided

- At times, the paper reads like a report that is too focused on the engineering details of the DOME environment. I would suggest moving some of these into an appendix and focusing on a few main problems that you are solving differently than the state of the art. Then show evidence that your novel solution is better than the state-of-the-art.

- I recommend clearly stating the contributions of the paper

Minor:

- Figure 2 caption: “Q-table converging after 10 minutes for one simulated Volvox” I am not sure whether the table has converged. Yes, all the agents have stopped after 5min, so the table doesn’t change much anymore. But that also means that the state part of the table containing low light visitation regions doesn’t get visited anymore.

- “The DOME operates on a Raspberry Pi computer, meaning that the reinforcement learning algorithms developed here, specifically Q-learning, can not be computationally expensive.“ Why could you not run the learning algorithm on an external computer that communicates to the Raspberry Pi?

- Consider changing writing from passive to active voice


**Summary Of The Paper:**

The authors propose learning control strategies in real-time for agent collectives.
They demonstrate the result of tabular Q-learning on a closed-loop Dynamic Optical Micro-Environment (DOME) platform to control the motion of light-responsive Volvox agents. Specifically, Q-learning allows learning how light may be projected onto Volvox algae to maximally reduce their velocity.

**Summary Of The Review:**

The paper describes the application of tabular Q-learning to the DOME environment. I am worried that the novelty and significance of the algorithmic contributions in this paper are too focused on the DOME environment to apply to other areas of learning. Competitive baselines, and a comparison to state-of-the-art are lacking as well.

---

> ### Author Response · Authors · 2021-11-22
> **Response to Reviewer QBsR**
>
> It is true that there is a considerable focus on the environment itself, rather than solely the learning algorithm. This is an understandable concern; however, we would argue that there is value in demonstrating the power of such a learning technique to address complex real-world systems, such as the biological agents used here. The revised version seeks to clarify this goal, and better highlight the motivation and contributions of the work. In particular, the motivation for developing an on-board, light-weight Q-learning algorithm for the limited computational capacity of the DOME is justified by the desire to maintain a self-contained system. This allows for potential future applications in which the system can be operated from within an environment such as a cell incubator or miniaturised into a wearable medical device.
>
> Regarding the absence of a comparisons to other competitive algorithms and baselines, this was not a focus as the primary goal of the work was to investigate if Q-learning could be applied to real time learning of Volvox dynamics. We would agree wholeheartedly however that this would be part of the next step in optimising the process, having found this approach to be effective.
>
> The code used in the simulator, as well as to control the Volvox, is available online and is now linked in the text.

---

### Decision · Program_Chairs · 2022-01-20

**Decision:**

Reject

**Comment:**

This paper applies and evaluates the use of Q-learning for the control of microscopic collectives of Volvox algae.

While the application is indeed very cool and potentially impactful, the paper has no theoretical contribution to the field of machine learning as it consists of an empirical evaluation of an existing (and well-established) algorithm.

The reviewers agree on the importance of the application, reported concerns about the current manuscript. In particular:
- Reviewers QBsR and GPp7 suggested including additional comparisons to other learning algorithms
- Reviewers QBsR and BtTc also suggested improving the writing

Overall, I agree with the reviewers that the current manuscript has a lot of potentials, but it could benefit from additional work.
Please carefully consider and incorporate the feedback received from the reviewers. Personally, I think that presenting a more sharp message and clearer insights would further increase the quality of exposition and help to make a stronger case for why this manuscript is relevant to the larger ML community.